# Gene Immunotherapy of Colon Carcinoma with IL-2 and IL-12 Using Gene Electrotransfer

**DOI:** 10.3390/ijms241612900

**Published:** 2023-08-17

**Authors:** Tilen Komel, Masa Omerzel, Urska Kamensek, Katarina Znidar, Ursa Lampreht Tratar, Simona Kranjc Brezar, Klemen Dolinar, Sergej Pirkmajer, Gregor Sersa, Maja Cemazar

**Affiliations:** 1Department of Experimental Oncology, Institute of Oncology Ljubljana, Zaloska 2, SI-1000 Ljubljana, Slovenia; tkomel@onko-i.si (T.K.); momerzel@onko-i.si (M.O.); ukamensek@onko-i.si (U.K.); kznidar@onko-i.si (K.Z.); ulampreht@onko-i.si (U.L.T.); skranjc@onko-i.si (S.K.B.); gsersa@onko-i.si (G.S.); 2Faculty of Medicine, University of Ljubljana, Vrazov trg 2, SI-1000 Ljubljana, Slovenia; 3Faculty of Medicine, Institute of Pathophysiology, University of Ljubljana, Zaloska 4, SI-1000 Ljubljana, Slovenia; klemen.dolinar@mf.uni-lj.si (K.D.); sergej.pirkmajer@mf.uni-lj.si (S.P.); 4Faculty of Health Sciences, University of Ljubljana, Zdravstvena pot 5, SI-1000 Ljubljana, Slovenia; 5Faculty of Health Sciences, University of Primorska, Polje 42, SI-6310 Izola, Slovenia

**Keywords:** CT26 colon carcinoma, gene electrotransfer, IL-2, IL-12, anti-tumour immune memory

## Abstract

Gene immunotherapy has become an important approach in the treatment of cancer. One example is the introduction of genes encoding immunostimulatory cytokines, such as interleukin 2 and interleukin 12, which stimulate immune cells in tumours. The aim of our study was to determine the effects of gene electrotransfer of plasmids encoding interleukin 2 and interleukin 12 individually and in combination in the CT26 murine colon carcinoma cell line in mice. In the in vitro experiment, the pulse protocol that resulted in the highest expression of IL-2 and IL-12 mRNA and proteins was used for the in vivo part. In vivo, tumour growth delay and also complete response were observed in the group treated with the plasmid combination. Compared to the control group, the highest levels of various immunostimulatory cytokines and increased immune infiltration were observed in the combination group. Long-term anti-tumour immunity was observed in the combination group after tumour re-challenge. In conclusion, our combination therapy efficiently eradicated CT26 colon carcinoma in mice and also generated strong anti-tumour immune memory.

## 1. Introduction

The immune system plays a key role in both the development and elimination of cancer. In recent years, immunotherapy has gained significant importance in the treatment of cancer, primarily through the use of checkpoint inhibitors [1]. However, some tumours do not respond to such treatment and research, focused on ways to improve cancer immunotherapy and boost the immune system, is needed.

One such area of research focuses on cytokines. To overcome the drawbacks of recombinant cytokine protein therapy used in the early 1990s, which resulted in unacceptable toxicity due to high cytokine doses, the gene therapy approach is currently being investigated. In gene therapy, nucleic acids are introduced into the patient’s cells, which then produce the therapeutic protein that can trigger an immune response against tumours [2]. There are several methods for delivery of nucleic acids, of which electroporation, a physical method, has already shown clinical applicability. Compared to viral vectors, electroporation shows great potential due to its low immunogenicity, low risk of insertional mutagenesis, and lower cost [3].

Electroporation is a delivery method based on the application of electrical pulses to cells and tissues to destabilize the cell membrane and facilitate the entry of molecules. When nucleic acids are introduced into the cells and tissues, the application is called gene electrotransfer [4,5]. Another biomedical application of electroporation, in which cytotoxic drugs are introduced into tumours, is called electrochemotherapy. Electrochemotherapy is one of the ablative therapies for the treatment of superficial and also deep-seated tumours of various histologies [6,7].

In the field of immunotherapy by gene transfer, most studies report electrotransfer of plasmid DNA encoding interleukin 12 (IL-12) [4,8,9]. Due to its potent pro-inflammatory functions IL-12 is a suitable candidate for cancer immunotherapy [10]. Alone or in combination with radiotherapy or chemotherapy, it shows good antitumour efficacy in a variety of tumours [11,12,13,14]. In addition, the combination with other types of immunotherapies, such as the cytokine tumour necrosis factor-α (TNF-α) or immune checkpoint inhibitors, has also been investigated [9,15,16]. Ongoing clinical studies are currently investigating the safety and efficacy of *Il-12* gene electrotransfer [17,18]. Among the various immunostimulatory cytokines, gene electrotransfer has also been evaluated for the administration of interleukin 2 (*Il-2*) or interleukin 15 (*Il-15*) [19,20]. 

We recently propose gene electrotransfer of *Il-12* in combination with *Il-2* to potentiate the antitumour effects of both interleukins [21]. The rationale for combining two cytokines is due to their mode of action. IL-12 induces interferon gamma (IFN-γ) production, which in turn stimulates infiltration of cytotoxic T lymphocytes and natural killer cells (NK cells) into the tumour [22]. Moreover, IFN-γ can induce increased MHC I expression on tumour cells, activates proinflammatory macrophages and it has also antiangiogenic effect [23]. It was demonstrated that local production of IL-2 by *Il-2* gene electrotransfer would cause clonal expansion, survival, proliferation, and differentiation of activated cytotoxic T lymphocytes [24]. In addition, these two cytokines have been shown to play an important role in the formation of trained immunity of innate immune cells such as macrophages, dendritic cells, and NK cells in combination with other proinflammatory interleukins [25,26]. After exposure to the first challenge, these cells may undergo long-term functional reprogramming, which then leads to an altered response to a second challenge after they have returned to an unactivated state [27]. Activation of both the innate and adaptive immune systems should therefore lead to an effective anti-tumour response and induction of a strong anti-tumour immune memory [28,29]. 

The aim of our current study was to investigate the efficacy of combined *Il-12* and *Il-2* gene transfer in the mouse colorectal carcinoma tumour model, which is considered an immunologically “hot “ tumour with many expressed neoantigens and high immune cell infiltration [30,31]. For this purpose, in vitro cell viability and expression of IL-12 and IL-2 were determined after gene electrotransfer. The in vivo efficacy of gene electrotransfer was determined by tumour growth delay, histological analysis of tumour sections, and cytokine expression profile. The abscopal effect was evaluated by measuring contralateral tumours, and an ex vivo cytotoxic assay was also performed to assess the activity of immune cells (splenocytes) after treatment. 

## 2. Results

### 2.1. In Vitro Survival after Electroporation

First, the survival of CT26 tumour cells was determined after treatment with two different pulse protocols alone. Treatment of cells with both pulse protocols alone resulted in a significant decrease in cell survival compared to the untreated control, by 25% in the EP2 and 60% in the EP1 group (Figure 1a) (*p* < 0.05). Treatment of cells with EP2 pulses alone resulted in significantly higher survival (more than twofold) compared to EP1 pulses (Figure 1a) (*p* < 0.05). In contrast, treatment of cells with three different plasmids alone (pIL-2, pIL-12, and pControl) and the combination of IL-2 and IL-12 plasmids (COMB) had no effect on cell survival (Figure 1b). Cell survival after electrotransfer of plasmids in combination with two different pulse protocols was also determined. After treatment with EP2 pulses, cell survival decreased equally (by approximately 25%) in all groups, regardless of the plasmid used (Figure 1d). The decrease was in the same range as in the cells exposed only to the EP2 pulse protocol. On the other hand, the survival rate of cells after treatment with EP1 pulses decreased equally by more than 80% in all groups, regardless of the plasmid used (Figure 1c). The survival rate was lower than that of cells exposed only to the EP1 pulse protocol. 

### 2.2. In Vitro Il-2 and Il-12 Transcripts Expression

Expression of *Il-2* and *Il-12* transcripts in CT26 cells was determined two days after gene electrotransfer in vitro. *Il-2* and *Il-12* transcripts were detected only in the corresponding groups after electrotransfer. The transcripts were not detected in the control group and in the groups to which only the plasmid was added. The levels of both transcripts were higher after the application of EP1 pulses than after EP2 pulses (Figure 2a,b) (*p* < 0.05). In addition, expression of the *Il-2* and *Il-12* transcripts was higher after electrotransfer with EP1 pulses in the group in which a single plasmid was added than in the group treated with the combination of both, although the differences were not significant.

### 2.3. In Vitro IL-2 and IL-12 Protein Expression

Expression of IL-2 and IL-12 proteins in cell culture media after gene electrotransfer was also determined by ELISA. As shown by the above mRNA transcripts expression results, IL-2 and IL-12 were detected only in the corresponding groups. The levels of both proteins were also higher after gene electrotransfer using EP1 pulses (Figure 3a, b). Protein expression of IL-2 (Figure 3b) and IL-12 (Figure 3a) transcripts was higher after gene electrotransfer with EP1 pulses in the group in which a single plasmid was added than in the combination group (*p* < 0.05). Due to the higher concentrations of both proteins, the EP1 pulse protocol was chosen for the in vivo experiments.

### 2.4. Therapeutic Effectiveness In Vivo

In the first in vivo experiment, tumour growth was monitored to determine the growth delay of tumours treated with the individual plasmids IL-2 and IL-12 alone and in combination. In all groups where electrotransfer of plasmids IL-2 (EP1 pIL-2), IL-12 (EP1 pIL-12), and their combination (EP1 COMB) was performed, a delay in tumour growth was observed compared to control groups (without treatment and treatment with EP or plasmids only). In the control groups, all tumours reached the humane endpoint within 8 to 10 days (Figure 4a). In the EP1 pIL-2 and EP1 pIL-12 groups, a low antitumour effect was observed; all mice reached the endpoint within 18 days (Figure 4b,c). In contrast, mice in the EP1 COMB group reached the experimental end point between 15 and 30 days (Figure 4d). The delay of tumour growth in the EP1 COMB group was significantly prolonged compared to all groups, except for the EP1 pIL-2 group (*p* < 0.05) (Figure 4e). Additionally, in this and in all of the following in vivo experiments we did not observe any changes in the behaviour of the mice after the treatment. Moreover, the body weight of the all the mice has not decreased during the time of the experiment. 

### 2.5. Transfection Efficiency In Vivo after Treatment Optimization with Collagenase and Hyaluronidase Pre-Treatment

Due to the low transfection efficiency and small antitumour effectiveness of therapy compared to previously published results [21], we aimed to increase transfection efficiency by pre-treatment of tumours with hyaluronidase and collagenase (C&H). To determine the effects of the enzymes on transfection efficiency, tumour cells were injected into both flanks of the mice. The tumour on the left flank was pre-treated with collagenase and hyaluronidase, whereas no enzymes were injected into the tumour on the right flank. Gene transfer was performed in both tumours with the control plasmid alone and a combination of the plasmids IL-2 and IL-12, and expression was measured 3 days later. As expected, increased expression of *Il-2* and *Il-12* mRNA transcripts was observed only in the groups treated with IL-2 and IL-12 plasmids (EP1 COMB and EP1 COMB C&H) (Figure 5). Moreover, expression was higher in the group pre-treated with the enzymes (EP1 COMB C&H) than in the non-pre-treated group (EP1 COMB).

### 2.6. Therapeutic Effectiveness In Vivo after Collagenase and Hyaluronidase Pre-Treatment

In the second in vivo experiment, tumours were pre-treated with collagenase and hyaluronidase before electrotransfer of the plasmids IL-2, IL-12 alone and of their combination. Tumour growth was subsequently monitored to determine the growth delay of the tumours after electrotransfer. Pre-treatment with these two enzymes prolonged growth delay in the EP1 pIL-2, EP1 pIL-12, and EP1 COMB groups, and complete responses were also obtained in the latter two groups. In the control groups, all mouse tumours reached the endpoint of the experiment within 8 to 10 days (Figure 6a), indicating that pre-treatment with the enzymes did not affect tumour growth in the control groups. Moreover, in the EP1 pIL-2 group, all mice reached the endpoint of the experiment within 22 days (Figure 6b). In contrast, in the EP1 pIL-12 group, a complete response was achieved in 2 out of 9 mice. In the remaining seven mice, tumours reached endpoint between 15 and 35 days after treatment (Figure 6c). Two mice in which a complete response was achieved received a re-injection of tumour cells on the opposite flank, and tumours developed within 19 days. In addition, in the EP1 COMB group, complete tumour response was observed in three mice, while in the remaining six mice, tumours also reached the experimental endpoint within 15 and 35 days (Figure 6d). In addition, three mice that had achieved a complete response were reinjected with tumour cells on the opposite flank, and one of them showed no tumour growth for a period of 100 days after reinjection. The efficacy of electrotransfer of the plasmid combination is also evident from the graph, in which tumour growth delay was highest compared to the other control groups and the EP1 pIL-2 group (*p* > 0.05) (Figure 6e). The growth delay in the EP1 pIL-12 and EP1 COMB groups was longer after pre-treatment with collagenase and hyaluronidase than without this pre-treatment, although the differences were not statistically significant. 

### 2.7. Abscopal Effect 

To determine the abscopal effect of treatment, two tumours were implanted, one on each flank of the mice, and the tumour on the right flank (primary) was treated. Electrotransfer of Control, IL-2 and IL-12 plasmids alone and their combination was performed to the tumours pre-treated with collagenase and hyaluronidase. Subsequently, the growth of primary and secondary tumours was observed. Similar to the previous experiment, a complete response of primary tumours was observed in 10% of mice in the EP1 pIL-12 group and in 30% in the EP1 COMB group. No complete response was observed in secondary tumours. In the EP1 pIL-2, EP1 pIL-12 and EP1 COMB groups, growth delay of secondary tumours or an abscopal effect was achieved. In the control groups, all mice reached the endpoint of the experiment within 8 to 9 days (Figure 7a). In addition, in the EP1 pIL-2 treatment group, all mice reached the end point of the experiment within 19 days (Figure 7b), in the EP1 pIL-12 group within 22 days (Figure 7c), and in the EP1 COMB group within 28 days, with one mouse reaching the end point after 50 days (Figure 7d). The efficacy of the treatment can also be seen in the growth delay graph (Figure 7e). A minimum growth delay of 2.9 days was achieved in the EP1 pIL-2 group, 6.7 days in the EP1 pIL-12 group, and 12.3 days in the EP1 COMB group. The growth delay in the EP1 COMB group was highest compared to all other groups (*p* < 0.05).

### 2.8. Ex Vivo Spleen Cytotoxicity Assay

To determine the cytotoxic activity of the immune cells of the treated animals, the ex vivo splenic cytotoxicity assay was performed. Electrotransfer of Control, IL-2, and IL-12 plasmids alone and their combination in CT26 was performed, splenocytes were isolated, and the assay was performed according to the procedure described in Materials and Methods. The specific survival of tumour cells at 48 h was lowest in the EP1 pIL-12 and EP1 COMB groups (Figure 8). In the EP1 COMB group, specific survival of tumour cells was 21%, which was statistically significantly lower than in all other groups except EP1 pIL-12 (*p* < 0.05). The differences between the other groups were not statistically significant.

### 2.9. Expression of Cytokines in Tumour

To determine the involvement of inflammatory and immune responses in the antitumour efficacy of treatment, the multiplex ELISA assay was used. In this way, the expression of specific cytokines involved in these responses was determined. The data are presented as a series of cytokines whose levels were increased in all groups treated by electroporation except EP1 pIL-2 (Figure 9) and were increased only in EP1 pIL-12 and EP1 COMB (Figure 10) compared with the untreated control group. In this way, we determined the effect of electroporation alone and the effect of IL-12 and IL-2 individually and in combination. The cytokines that were increased in all electroporation-treated groups were eotaxin, granulocyte–macrophage colony-stimulating factor (GM-CSF), IL-15, IL-17, IL-6, IP-10 (CXCL10), MIP-1α, MIP-1β, and MIP-2 (Figure 9). The levels of eotaxin, GM-CSF, IL-17, IL-6, and IP-10 were also higher in the EP1 COMB group, although the differences were not statistically significant compared with the other electroporated groups (*p* > 0.05). Interleukin 12 (IL-12 (p70)) levels were higher in the EP1 COMB group compared to all other groups (Figure 10) (*p* < 0.05). However, in the EP1 pIL-12 group, this difference was only observed compared to the CTRL and EP1 pIL-2 groups (Figure 10). IFN-γ levels were also highest in the EP1 COMB group and second highest in the EP1 pIL-12 group (Figure 10) (*p* < 0.05). The same trend was observed for TNF-α levels, with statistically significant differences only between the CTRL and EP1 COMB groups (*p* < 0.05). There was no difference in IL-2 levels between treatment groups.

### 2.10. Immunohistochemical Evaluation of Immune Cell Infiltration in Tumours and Vascularisation after Combined Treatment

To evaluate the efficacy of the combined treatment in tumour regression and immune infiltration, histological analysis of tumour sections was performed 3 days after treatment. Haematoxylin and eosin (H&E) staining was performed to evaluate the effects of the different treatments on tumour tissue by distinguishing necrotic areas from viable areas (Figure 11a). The results show that the electroporation-treated tumours have a larger necrotic area than the non-electroporated ones (Figure 11b).

Immunohistochemical staining was used to determine immune cell infiltration and assessment of innate and adaptive immune responses. Anti-F4/80 and anti-MHC II antibodies specific for macrophages and M1 macrophages and anti-CD11c antibodies specific for dendritic cells were used to assess the innate immune response (Figure 12). Anti-CD4 and anti-CD8 antibodies specific for CD4+ helper and Cytotoxic T lymphocytes, respectively, were used to determine the adaptive immune response (Figure 13).

Staining for F4/80 showed many positive cells in the tumour of EP1 pIL-2, EP1 pIL-12, EP1 COMB, and SKIN LPS groups (positive control), whereas no or very few cells were observed in the skin or tumour of SKIN CTRL (negative control) and CTRL, EP1, pControl, pIL-2, pIL-12, and EP1 pControl groups (Figure 12a). The SKIN LPS control had the largest number of positive cells compared to all other groups (*p* < 0.05). However, the number of positive cells in the tumours of EP1 pIL-2, EP1 pIL-12, and EP1 COMB groups was greater than in the tumours of the other electroporated and non-electroporated control groups (*p* < 0.05) (Figure 12b). In contrast, staining for MHC II showed a high percentage of positive cells only in the tumours of EP1 pIL-12 and EP1 COMB groups (Figure 12c). The percentage of positive cells in the tumours of the EP1 pIL-12 group was higher than in the tumours of the other electroporated and non-electroporated control groups (*p* < 0.05), while the percentage of positive cells in the tumours of the EP1 COMB group was higher than in all other groups (*p* < 0.05). The percentage of positive cells in the tumours of the EP1 pIL-2 group was about the same as in the tumours of the other control groups. Very few MHC II positive cells were detected in the skin of the SKIN CTRL group and also in the SKIN LPS group. Staining for CD11c also showed many positive cells in the tumours or skin of the EP1 pIL-2, EP1 pIL-12, EP1 COMB, and SKIN LPS groups, while none or very few cells were observed in the skin of the SKIN CTRL group. A small accumulation of CD11c+ was observed in the tumours of groups CTRL, EP1, pControl, pIL-2, pIL-12, and EP1 pControl (Figure 12a). However, the number of positive cells was higher in the tumours or skin of EP1 pIL-2, EP1 pIL-12, EP1 COMB, and SKIN LPS groups than in the tumours of the other electroporated and nonelectroporated control groups (*p* < 0.05) (Figure 12d).

Staining for CD4 showed a large number of positive cells in the tumours of EP1 pIL-2, EP1 pIL-12, EP1 COMB, and SKIN LPS groups (Figure 13a). However, the difference in the number of positive cells in the tumours of EP1 pIL-2 and EP1 pIL-12 compared with the other electroporated and nonelectroporated control groups was not statistically significant (Figure 13b). In the skin of the SKIN LPS group, the number of positive cells was greater compared with the other electroporated and nonelectroporated control groups; conversely, the number of positive cells was greater in the EP1 COMB group compared with all groups (*p* < 0.05) (Figure 13b). Very few or almost no cells were observed in the SKIN CTRL group. Staining for CD8 showed a large number of positive cells in groups EP1 pIL-12 and EP1 COMB (Figure 13a). The number of positive cells was greater in the EP1 pIL-12 group than in the other electroporated and non-electroporated control groups; conversely, the number of positive cells was greater in the EP1 COMB group than in all groups (*p* < 0.05) (Figure 13c). The number of positive cells in EP1 pIL-2 was approximately the same compared to the other electroporated and non-electroporated control groups (Figure 13c). Very few CD8-positive cells were detected in the SKIN CTRL group and also in the SKIN LPS group.

In addition, IHC staining of tumour sections with anti-CD31 antibodies specific for endothelial cells was performed to determine whether the combination treatment affected tumour vasculature and, in particular, the number of vessels present in the tumour. The number of vessels was lowest in the tumours of EP1 pIL-2, EP1 pIL-12, and EP1 COMB (Figure 14a). The number of vessels in the tumours of EP1 pIL-2 and EP1 pIL-12 groups was the lowest compared with the electroporated and nonelectroporated control groups. Conversely, the number of vessels was lower in the tumours of group EP1 COMB compared with the tumours of all groups except EP1 pIL-12 (Figure 14b) (*p* < 0.05).

## 3. Discussion

In this study, the antitumour and antiangiogenic activity of a gene-electrotransfer combination of plasmids with transcripts for interleukins IL-2 and IL-12 was investigated in the CT26 mouse tumour model. After gene electrotransfer, tumour growth delay was observed in all groups treated with IL-2 and IL-12 individually and in combination, which was highest in the combination group. Since complete response was not observed in any of the groups, tumours were pretreated with collagenase and hyaluronidase before gene electrotransfer. Tumour growth delay and complete response were subsequently observed in the EP1 pIL-12 and EP1 COMB groups, with the latter group having the greatest growth delay and the highest percentage of complete responses. In the tumours of the EP1 COMB group, the infiltration of immune cells and the concentration of various proinflammatory cytokines were the highest. The antitumour effect in the EP1 COMB group was also characterised by the enhanced ex vivo cytotoxic activity of splenocytes and the occurrence of the abscopal effect.

Several studies suggest that pro-inflammatory interleukins such as IL-2 and IL-12 play an important role in the activation and targeting of cytotoxic T lymphocytes and other immune cells to the tumour microenvironment [32,33]. However, therapy with recombinant IL-2 or IL-12 protein has many side effects because of the mode of administration and the high doses required for a successful response to therapy. Gene transfer is one of the more effective and safer ways to deliver DNA molecules directly into the tumour or surrounding tissue. The antitumour efficacy of gene transfer of single pIL-2 or pIL-12 molecules has been demonstrated in a number of preclinical and clinical studies [20,34].

First, the cytotoxicity of plasmids encoding IL-2 and IL-12 administered with EP1 and EP2 pulse protocols in vitro was investigated. These pulse protocols or similar protocols are used as standard protocols in many gene therapy studies by our and other research groups [19]. Survival of cells treated with the EP1 pulse protocol was significantly reduced, whereas treatment of cells with the EP2 pulse protocol resulted in a small reduction in viability. Moreover, cell viability decreased significantly after the addition of plasmids to cells exposed to the EP1 pulse protocol, whereas the addition of plasmids to cells exposed to the EP2 pulse protocol had little effect on cell viability. The results of the cytotoxicity of the plasmids are comparable to the results of our previous study and also to other studies [21,35].

Second, transfection efficiency was determined in vitro 48 h after electroporation by measuring the mRNA and protein levels of IL-2 and IL-12. Both interleukin transcripts and their protein forms were detected only in the groups treated with the corresponding plasmid in combination with electroporation. As already observed in our previous study on B16F10 cells, transcript expression and protein levels in CT26 cells were higher after electroporation with EP1 pulses than after EP2 pulses [21]. However, the expression observed in CT26 cells was lower than in B16F10 cells.

Based on the in vitro results and our previous in vivo study on B16F10 tumour model the EP1 pulse protocol was selected for further in vivo experiments [21]. A number of studies have investigated IL-12 gene electrotransfer as monotherapy and demonstrated its efficacy in a range of histologically diverse tumours [36,37,38]. Beside studies that investigated viral [39] and non-viral [40] local gene delivery of gene IL-12, the effectiveness of gene electrotransfer of this gene in tumours was demonstrated in our previous studies [9,21]. In our study, despite a significant delay in tumour growth after gene electrotransfer of combined pIL-2 and pIL-12, no complete response was observed. Therefore, we decided to optimise the treatment to increase the transfection efficiency. The in vivo tissue organisation and the composition of the extracellular matrix (collagen fibres, proteoglycans, glycosaminoglycans, etc.) make it difficult for the plasmid DNA to move and spread in the tissue. Therefore, the expression level of the transgene in tumours is low and never reaches more than a few percentage points [41]. To improve the distribution of plasmid DNA in the tumour before gene electrotransfer, tumours were pre-treated with hyaluronidase (an enzyme that degrades hyaluronan, a component of the extracellular matrix) and collagenase (an enzyme that degrades collagen). This has been shown to increase transgene expression up to 30-fold in tumours with high extracellular matrix content [42]. Similar observations were made in this study, in which an increase in mRNA expression of the IL-2 and IL-12 transcripts was observed after optimised gene electrotransfer. Therefore, in vivo gene electrotransfer was again performed in CT26 tumours, but this time the tumours were pretreated with collagenase and hyaluronidase. In EP1 pIL-12 and EP1 COMB groups, significant tumour growth delay was achieved compared to other electroporated and non-electroporated controls. In addition, complete responses were observed in 20% of mice in the EP1 pIL-12 group and in 30% in the EP1 COMB group. 

An abscopal effect was also observed in the groups treated with pIL-2 and pIL-12, as well as in the combination of the two, with the growth delay in the latter group. The abscopal effect occurs when shrinkage of the untreated tumours occurs simultaneously with shrinkage of the treated tumours (with immunotherapy to enhance the abscopal effect, nature). A strong immune response is required for an abscopal effect to occur. The key event is the activation of T lymphocytes, which occurs through the release of tumour antigens, the release of damage-associated molecular patterns (DAMPs), the uptake and processing of tumour antigens by APC), the presentation of antigens by APCs to naive T lymphocytes, and the activation and proliferation of tumour-specific cytotoxic and helper T lymphocytes [43]. A similar observation of the abscopal effect was also described in a study combining electrochemotherapy with pIL-12 gene transfer into the tumour environment [11]. In addition, immune cell activity was also tested after optimised combination therapy in the CT26 model. Cytotoxic T lymphocytes are the most potent effectors in the immune response against cancer and form the basis for successful cancer immunotherapies [44]. The activity of splenocytes isolated from the spleens of mice in the EP1 COMB group was higher than that of the EP1 pIL-2 and EP1 pIL-12 groups and the control groups. In this way, we were able to further demonstrate the efficacy of our combination therapy.

To identify the mechanisms of antitumour efficacy in the CT26 model after optimised therapy, the levels of specific cytokines involved in an inflammatory or immune response were determined. Most of the cytokines that were elevated in the tumours of all electroporation-treated groups compared with the tumours of the nonelectroporated control are directly or indirectly involved in the activation of the innate immune response. Electrical pulses with lower voltage but longer duration, such as the EP1 protocol, can cause irreversible electroporation and thus membrane perforation that cannot be closed. Furthermore, this may lead to a massive release of intracellular hidden tumour antigens, triggering a potential antitumour immune response and causing infiltration of immune cells [45,46]. However, the addition of potent proinflammatory cytokines such as IL-2 and IL-12 with strong immunomodulatory properties may enhance the immune response elicited by electroporation. 

In the EP1 COMB group, the levels of eotaxin and IL-17 were increased. These two cytokines are considered eosinophil chemoattractants involved in the recruitment of inflammatory cells such as eosinophils and neutrophils, and are positively related to the neutrophil count in the tumour microenvironment [47,48]. The presence of eosinophils either in the tumour or in the peripheral blood is a favourable prognostic factor for most cancers [49]. Moreover, in patients with colorectal carcinoma, the increased number of CD15+ neutrophils correlated positively with patient survival [50]. Another cytokine that showed increased levels in the treated tumours was GM-CSF. It is a cytokine that regulates proliferation and differentiation of myeloid stem cells and mature granulocytes. It also enhances the inflammatory response by promoting the activation of various types of immune cells [51]. GM-CSF is used as an immunostimulant in cancer therapy, as an adjuvant in vaccines, and the results of the clinical trial have shown that GM-CSF-based gene therapy is an ideal approach to improve treatment outcomes in melanoma [52]. Elevated levels of IP-10 and IL-6 were also observed. These two cytokines play an important role in the activation of cells belonging to the adaptive immune response [53,54]. IP-10 is secreted by a variety of cells including leukocytes, activated neutrophils, eosinophils, monocytes, stromal, endothelial, and epithelial cells in response to IFN-γ. It is also a chemoattractant for monocytes, T lymphocytes and NK cells [55]. Previous studies have shown that the presence of IP-10, among other chemokines, correlates with different subsets of immune cells and with high densities of T lymphocyte subpopulations within specific tumour regions. High expression of these molecules correlated with prolonged disease-free survival [56]. IL-6, on the other hand, is a multifunctional cytokine that regulates immunity, is mainly produced by T lymphocytes, and acts on activated B cells, initiating their final maturation into antibody-producing cells. In addition, IL-6 can induce proliferation and differentiation of cytotoxic T lymphocytes in the presence of IL-2 [53]. IL-6 also down-regulates T lymphocyte apoptosis and maintains anti-tumour immunity in the lymph nodes, where lymphocyte priming takes place, and in the tumour microenvironment, where IL-6 promotes the recruitment of effector T lymphocytes [57]. The increased concentrations of these cytokines may link the effect of tumour electroporation to activation of the innate anti-tumour immune response. However, properties that promote tumour growth have also been observed in most of these cytokines. Our results suggest that the addition of IL-2 and IL-12 or their combination may shift the balance in favour of tumour eradication. 

Lastly, elevated concentrations of IL-12, IFN-γ and TNF-α were found only in groups where pIL-12 was added. The increased IL-12 levels after intratumoural gene electrotransfer were confirmed in our previous [21] and also in other studies [58]. IL-12 concentrations were higher in the EP1 COMB group than in the monotherapy group, which may be the result of endogenous expression caused by functional cooperation between them, leading to a positive feedback loop. In addition to the increased IL-12 levels in these groups, IFN-γ levels were also increased, which is the result of increased production by helper T lymphocytes stimulated by IL-12. The remarkable therapeutic efficacy of treatment with the plasmid combination compared with monotherapies could also be explained by the increased levels of these two cytokines. Finally, increased TNF-α levels were also observed in the EP1 COMB group. TNF-α is mainly produced by activated macrophages, T lymphocytes, and NK cells. Its anti-tumour effect is now well established and may be mediated by a variety of mechanisms, including activation of T effector lymphocytes (including macrophages and NK cells), inducing tumour microvasculature collapse by modulating endothelial cells and disrupting neoangiogenesis, and promoting tumour-associated macrophages (TAM) to the M1 anti-tumour stage, which suppress tumour cell growth in the tumour microenvironment [59]. TNF-α is also involved in attracting and stimulating neutrophils and monocytes to activation sites for anti-tumour immune responses [60]. Tumours can generally be divided to immunologically “cold “ tumours, with low levels of cytokine production and T cells infiltration, and to “hot “ tumours with high levels of cytokine production and T cells infiltration [31]. CT26 tumour is considered an immunological hot tumour that usually responds well to immunotherapy. In our case, the effect of combined cytokine therapy was not as effective as it had been previously in the immunologically “cold” tumour B16F10 [21]. This finding may be attributed to lower transfection efficiency after gene electrotransfer of pIL-2 and pIL-12 in CT26 tumours and consequently lower expression of interleukins. However, in the current study, we demonstrated that treatment optimization can improve the efficacy of therapy.

The antitumour mechanisms were also evaluated by histological and immunohistological analyses. HE staining showed larger necrotic areas in all electroporated groups. These results are also consistent with Magpix results, in which increased levels of some proinflammatory cytokines were observed in all electroporated groups. The electrical pulses caused tumour cells to die and antigens to be released, which in turn activated the immune system [46]. Furthermore, 3 days after treatment increased infiltration of dendritic cells, macrophages, and lymphocytes was determined in EP1 pIL-2, EP1 pIL-12, and EP1 COMB groups. Positive staining for F4/80 was observed in the tumours of all groups. Higher signals were observed in EP1 pIL-2, EP1 pIL-12, and EP1 COMB groups compared with other control tumours. As with F4/80, positive staining was observed in MHC II in all groups, and minimal signal was also observed in the SKIN CTRL group. The MHC class II is mainly expressed by APCs such as dendritic cells, monocytes, and macrophages. However, it has been demonstrated that nonprofessional APG (mast cells, lymphoid and vascular endothelial cells, and neutrophils) are also capable of inducible expression of MHC class II [61]. This could explain the more intense MHC II staining in the treatment groups compared with F4/80 staining. The same trend as for F4/80 staining was also observed for CD11c staining. Increased cell numbers were observed in EP1 pIL-2, EP1 pIL-12, and EP1 COMB groups, whereas none were observed in SKIN CTRL. Like macrophages, dendritic cells contribute to tumour cell killing by presenting antigens to cytotoxic and helper T lymphocytes. When both cell types are stimulated, they also produce large amounts of IL-12, which in turn activates T lymphocytes and NK cells [38,62]. This can be seen from the increased IL-12 levels (Figure 10) in the EP1 pIL-12 and EP1 COMB groups, which is also a consequence of a positive feedback loop. The presence of cytotoxic and helper T lymphocytes was also confirmed. They were observed in all groups except SKIN CTRL, but the highest infiltration of both types of T lymphocytes was observed in the EP1 COMB group. In the EP1 pIL-12 group, only cytotoxic T lymphocytes were statistically increased. Finally, the number of vessels in tumour sections stained with CD31 antibody was determined. The lowest number of vessels compared with the control groups was observed in the EP1 pIL-2, EP1 pIL-12, and EP1 COMB groups, whereas there were no differences in the number of vessels between the control groups. Overexpression of IL-12 has been shown to regulate tumour vasculature by inhibiting angiogenesis in an IFN-γ-dependent manner. The inhibitory effects of IL-12 on tumour vasculature were also associated with increased levels of the IFN-γ-inducible chemokine ligands CXCL9 and CXCL10 (IP-10) and decreased production of vascular endothelial growth factor (VEGF) [38]. This can also be confirmed by the increased IP-10 levels in the EP1 COMB group detected by Magpix assay. However, the VEGF levels remained the same in all groups. It is worth mentioning, that VEGF plays an important role in malignant disease progression as its overexpression in tumours is linked with increased proliferation and metastasis of the tumour cells [63]. Nevertheless, the increased levels of VEGF are not only problematic in cancer but also in other inflammatory diseases as it has a significant impact in the immune cell activation and infiltration to the tissue [64]. In the study of Gomułka et al. it was noticed that the elevated VEGF levels in asthmatic patients are positively correlated with the activation of basophils and thus influence the course of the disease [65]. 

In conclusion, our data suggest a combined role of IL-2 and IL-12 proteins in the recruitment of various types of immune cells such as dendritic cells, proinflammatory M1 macrophages, and T lymphocytes, the enhanced expression of various proinflammatory cytokines, and also in vascular destruction leading to tumour eradication. The results show that the combination of the two proteins can efficiently stimulate both adaptive and innate immune responses. In addition, the electrical pulses themselves may play an important role in activating the innate immune system by releasing tumour antigens into the tumour environment, which, together with the expression of proinflammatory cytokines such as GM-CSF, IL-15, and TNF-α, may promote innate trained immunity [66]. However, our study has some limitations, especially in determining additional classes of immune cells. In addition, larger groups of in vivo experiments may be required to obtain statistically significant differences in tumour growth delay and cytokine expression levels.

## 4. Materials and Methods

### 4.1. Cell Lines

The CT26 murine colon carcinoma cells (American Type Culture Collection, Manassas, VA, USA) were cultured in an advanced minimum essential medium (RPMI 1640; Gibco, Thermo Fisher Scientific, Waltham, MA, USA), supplemented with 5% (*v*/*v*) foetal bovine serum (FBS; Gibco), 10 mL/L L-glutamine (GlutaMAX; Gibco), 10 mL/L Penicillin–Streptomycin (stock solution, 10,000 U/mL, Gibco) in a 5% CO_2_ humidified incubator at 37 °C. The cells were routinely tested and confirmed to be free from mycoplasma infection, using MycoAlertTM PLUS Mycoplasma Detection Kit (Lonza Group Ltd., Basel, Switzerland).

### 4.2. Plasmids

Three different plasmids were used in the experiments; pORFmIL-12 (p40 p35) (InvivoGen, Toulouse, France) coding for mouse IL-12, pUNO1-mIL02 (InvivoGen) coding for mouse IL-2 and the pControl [67] used as a control plasmid.

All the plasmids were amplified in competent E. coli and purified using Endo Free Plasmid Mega Kits (Qiagen, Hilden, Germany) according to the manufacturer’s protocol and dissolved in endotoxin-free water at a concentration of 1 mg/mL or 0.5 mg/mL. The identity of plasmids was verified by restriction analysis and subsequent agarose gel electrophoresis. The purity of isolated plasmids was also determined using Cytation 1 multimodal reader with Take3™ Micro-Volume Plate (BioTek, Winooski, VT, USA) measuring the 260/280 and 260/230 absorbance ratio. The concentration of plasmid was measured using Qubit 3.0 Fluorometer (Thermo Fisher Scientific). The results of the plasmid`s purity and restriction are already published in our previous study [21]. 

### 4.3. In Vitro Electroporation Experiments

The in vitro electroporation protocol and procedure in CT26 cells in this study was performed as already described in our previous study in B16F10 cells. The cell viability, mRNA and protein expression after the gene electrotransfer were analysed using the reagents (VWR, Vienna, Austria and Thermo Fisher Scientific), primers (Integrated DNA Technologies, Coralville, IA, USA) and ELISA kits from the same provider (Thermo Fisher Scientific) and according to the manufacturer’s protocols [21].

### 4.4. Animals and Tumour Induction

In the experiment, 6–8 weeks-old female BALB/c mice (Charles River, Lecco, Italy) were used. The CT26 murine colon carcinoma cells were resuspended in saline solution in the concentration of 3 × 10^6^ cells/mL and 100 μL was injected subcutaneously into the shaved right flank of mice, i.e., primary tumours (tumours that were treated with plasmids coupled with electroporation). For monitoring the abscopal effect 100 μL of CT26 cells were injected subcutaneously into the shaved left flank of mice at the concentration of 3 × 10^5^ cells/mL (i.e., secondary tumours) at the same time as primary tumours. Six mice per cage were housed in specific pathogen-free conditions in a carousel mouse IVC rack system (Animal Care Systems Inc., Centennial, CO, USA) at a relative humidity of 55 ± 10%, a temperature of 20–24 °C and a 12 h light/dark cycle. Food and water were provided ad libitum. The mice were checked every day to assess their wellbeing. They were monitored for any unusual behaviour and for the changes of their grimaces [68]. Moreover, the mice body weight was also measured three times per week. The experimental procedures were performed in compliance with the guidelines for animal experiments of the EU directive (2010/63/EU) and the permission from the Veterinary Administration of the Ministry of Agriculture, Forestry and Food of the Republic of Slovenia (permission no. U34401-35/2020/8 and U34401-3/2022/11).

### 4.5. In Vivo Electroporation

When the primary tumours reached a volume of approximately 50 mm^3^ mice were randomly distributed in experimental groups. Then EP1 pulses were delivered 10 min after intratumoural (i.t.) injection of 50 µL of plasmid DNA (0.5 mg/mL) or endotoxin-free water. The pulses were generated using an electric pulse generator (ELECTRO cell B10, Betatech, Saint-Orens-de-Gameville, France) and delivered in 2 sets of 4 pulses in perpendicular directions through 2 parallel stainless-steel electrodes with a 6 mm gap between them. The water-based gel was applied to ensure good conductivity at the contact between the tumour and the electrodes (Ultragel, Budapest, Hungary). During the entire procedure, the animals were kept under inhalation anaesthesia with 1.5% isoflurane (Izofluran Torrex para 250 mL, Chiesi Slovenia, Ljubljana, Slovenia) delivered with an oxygen concentrator (Supera Anesthesia Innovations, Estacada, OR, USA) at a flow rate of 1 L/min. The tumours were pre-treated with collagenase and hyaluronidase prior to the delivery of electric pulses. The rationale for pre-treatment was to increase the transfection efficiency of gene electrotransfer in solid subcutaneous tumours with a high extracellular matrix content. The pre-treatment should lead to extracellular matrix degradation, allowing more efficient distribution of plasmid DNA [69]. The time points for the injection of each of the enzymes were selected according to the literature and were 24 h before the electroporation for collagenase and 2 h before the electroporation for hyaluronidase [42]. Collagenase was obtained from Worthington Biochemical (Worthington Biochemical Corp., NJ, USA) and dissolved in 0.9% NaCl and injected intratumourally at a concentration of 30 μg/50 μL per tumour. Hyaluronidase was obtained from Sigma-Aldrich (Sigma-Aldrich, St. Louis, MO, USA) and dissolved in 0.9% NaCl and injected intratumourally at a concentration of 500 IU/50 μL per tumour. 

Experimental groups were: the injection of endotoxin-free water alone (negative Control) and in combination with electroporation (EP1), the injection of individual plasmids alone and the combination of pIL-12 + pIL-2 plasmids (ratio of 1:1, 25 µL each) coupled with EP1 protocol (EP1 pControl, EP1 pIL-12, EP1 pIL-2, and EP1 COMB), without electroporation (pControl, pIL-12, and pIL-2) and the injection of lipopolysaccharide (Sigma-Aldrich) (SKIN LPS).

### 4.6. Tumour Growth Measurement

After the therapy, tumour growth was monitored three times per week using a Vernier calliper to measure the tumour volume. The volumes were calculated using the following equation: V = a × b × c × π/6, where a, b, and c represent the perpendicular diameters of the tumour. The animals were euthanized when the primary or secondary tumour reached predetermined humane endpoint of a volume of 350 mm^3^. The mice that had complete response (tumour-free for 100 days) were re-challenged with subcutaneous injection of 3 × 10^5^ CT26 cells resuspended in 100 µL saline solution in the mice’s left flank and monitored for tumour outgrowth. If the tumours did not develop, the mice were followed for another 100 days. Tumour growth delay was determined as the difference in time when treated and untreated tumours reached a tumour volume of 200 mm^3^. The mice with complete response were assigned with a growth delay of 60 days.

### 4.7. Ex Vivo Spleen Cytotoxicity Assay

Spleens were aseptically harvested from CT26 tumour-bearing mice (n=3 and 4 mice per group) 7 days after the start of the treatments. The spleens were disintegrated using a 50 μm sterile strainer (Sysmex, Norderstedt, Germany) and the splenocytes were suspended in 10 mL of cooled phosphate buffer solution (PBS, 4 °C). The cell suspension was centrifuged at 400× *g*, 4 °C for 5 min and the pellet was washed with cooled PBS (4 °C). Afterwards, the pellet was resuspended in red blood cell lysis buffer (BioLegend, San Diego, CA, USA) and incubated on ice for 5 min. The reaction was stopped by the 4× dilution with PBS. The splenocytes were then centrifuged at 400× *g*, 4 °C for 5 min, resuspended in 10 mL RPMI hepes (Sigma-Aldrich) and rested for 2h in a 5% CO2 humidified incubator at 37 °C before the assay.

CT26 cells were fluorescently labelled for the cytotoxicity assay using CFSE Cell Division Tracker Kit (BioLegend) according to manufacturer instructions. Then 100 μL of the labelled cells (target) at a concentration of 1.5 × 10^4^ cells/mL, were seeded on 96 well plates. After 24 h, freshly isolated splenocytes (effector) were added to labelled CT26 cells at a target: effector ratio of 1:10. The co-culture of target and effector cells were then incubated for 48 h in a 5% CO_2_ humidified incubator at 37 °C. After 48 h the cell media was removed and the wells were washed with PBS. The number of fluorescently labelled tumour cells that survived was determined with Cytation 1 multimodal reader. Images were taken using a 4 × objective lens and GFP (469/525) imaging filter cube for the detection of labelled tumour cells. Each well was focused using a laser autofocus cube and camera exposure settings were optimized manually. The images were processed with a black background and a 100 μm rolling ball, which were used to remove background fluorescence. Images were then analysed by masking the fluorescently labelled tumour cells and thus determining the number of tumour cells in co-culture and control wells after the washing step. The specific survival was calculated using the following formula.
$$specific\;survival\;\left[\%\right]=\frac{number\;ofCFSEpos\;tumour\;cells\;in\;co-culture}{number\;of\;CFSEpos\;tumour\;cells\;in\;control\;wells\;}$$

### 4.8. Multiplex Assay

For the in vivo part of the experiment, a multiplex ELISA assay was performed on tumours, taken on day 3 after the treatment. The tumours were excised and one half of the tumour was snap frozen in liquid nitrogen and stored at −80 °C. Frozen tumours were afterwards grinded to a fine powder under liquid nitrogen in a mortar with a pestle. The homogenized tumours were weighted and resuspended in 500 µL PBS and Protease Cocktail (Thermo Fisher Scientific). The suspension was then centrifuged for 10 min at 3000× *g* and the supernatant was collected and stored at −80 °C until further analysis. The concentration of 32 cytokines was measured using a mouse cytokine/chemokine magnetic bead panel (MCYTMAG-70K-PX32) (Merck KGaA, Darmstadt, Germany) on the MagPix instrument (Luminex, Austin, TX, USA) according to the manufacturer’s instructions. The concentration data were normalized to tumour weight. 

### 4.9. Histological Analysis

The other half of the tumours harvested on day 3 after the treatment were fixed for 48h in 10% neutral buffered formalin (Sigma-Aldrich) and afterwards embedded in paraffin. Additionally, positive and negative skin control samples were obtained from non-tumour-bearing mice. For positive skin control an injection of 25 µL of LPS (Sigma-Aldrich) at a concentration of 2 mg/mL was injected intradermal on the right flank of the mice. The skin from the area of LPS injection was excised 3 days after the LPS injection as a positive control. The skin from the area on the opposite flank was also excised as negative control. The samples were left overnight in a 10% buffered formalin (BD Biosciences, San José, CA, USA) and then transferred into 70% ethanol (Sigma-Aldrich). Subsequently, the tumours and skin were embedded in paraffin. The paraffin-embedded blocks were cut in 4 μm thick sections for immunohistochemical staining and 2 µm thick sections for H&E, one section per tumour. The latter were stained using H&E according to standard procedures. Necrotic areas, which appeared pink coloured, as well as the viable purple parts were determined. Images of H&E-stained tumours were obtained using a DP72 CCD camera connected to a BX-51 microscope (Olympus Corporation, Tokyo, Japan) at 4× magnification. The percent of necrotic tissue in a tumour section was assessed in a blind fashion by 3 examiners.

### 4.10. Immunohistochemistry

Immunohistochemistry was performed on 4-µm-thick paraffin tissue sections, using 6 different primary antibodies, detecting the presence of macrophages (anti-F4/80 Ab, anti-MHCII Ab), dendritic cells (anti-CD11c Ab), helper T lymphocytes (anti-CD4 Ab), cytotoxic T lymphocytes (anti-CD8 Ab), and endothelial cells (anti-CD31 Ab). Before the use of anti-F4/80, anti-CD11c, anti-CD4, anti-CD8, and anti-CD31 antibodies the antigen retrieval procedure was performed in a sodium citrate buffer (10 mM sodium citrate, pH 6.0), for 30 minutes at 95 °C. For anti-MHCII antibodies, no antigen retrieval procedure was used. Afterwards, sections were incubated with the primary antibodies in a humid chamber overnight at 4 °C. The following primary antibody dilutions in blocking buffer were used: 1/100 for rabbit anti-mouse F4/80 (ab111101), 1/200 for rabbit anti-mouse MHC class II (ab180779), 1/100 for rabbit anti-mouse CD4 (ab183685), 1/1000 for rabbit anti-mouse CD8 (ab209775), 1/200 for rabbit anti-mouse CD31 (ab28364) (Abcam, Cambridge, UK) and 1/200 for rabbit anti-mouse CD11c (D1V9Y) (Cell Signalling Technology, Danvers, MA, USA). Incubation with the primary antibody was omitted for the negative controls. For positive control samples, skins were taken from mice that had been injected subcutaneously with LPS. For these sections rabbit specific HRP/DAB (ABC) Detection IHC Kit (ab64261) was used. The staining was performed according to the manufacturer’s protocol. For the nuclear labelling haematoxylin counterstaining was employed. The immunostained slides were observed under bright light by an Olympus BX-51 microscope (Olympus Corporation) connected to a DP72 CCD camera. The images were captured at 40× magnification. For each tumour and skin section, four image fields were taken, 2 at the edge of a tumour and 2 at the centre. The average numbers of dendritic cells, helper T lymphocytes and cytotoxic T lymphocytes (were assessed in a blind fashion by 3 examiners. To quantify the infiltration of macrophages the percentage of positive cells in relation to the total number of cells in the field of view was determined in a blind fashion by 3 examiners. To measure the number of the blood vessels, binary masks were created from bright images taken at 40× magnification. The original image was processed with a low-pass filter and the RGB threshold was determined to distinguish between the blood vessels and the surrounding tissue (the pixels belonging to the blood vessels were given a value of 1, all other pixels below the threshold were given a value of 0). Binary masks of the detected blood vessels were then superimposed on the original image and discrepancies between the blood vessel network on the original image and the binary mask were manually corrected. Using the corrected binary masks, the number of the blood vessels was determined using image analysis software 4.8.2. (AxioVision, CarlZeiss, Jena, Germany).

### 4.11. Statistical Analysis

All data were tested for normal distribution using the Shapiro–Wilk test. Differences between experimental groups in the experiments were analysed using one-way analysis of variance (ANOVA), followed by Tukey’s tests and Student’s t-tests for group comparisons. If the data were not normally distributed, differences between experimental groups were analysed using Kruskal–Wallis tests, followed by Dunn’s and Mann–Whitney tests for group comparisons. GraphPad Prism software 9.5.0. (GraphPad Software, Boston, MA, USA) was used for graphing and other statistical analyses. A p-value of less than 0.05 was considered significant. Data are presented as arithmetic mean (AM) and standard error (SE).

## 5. Conclusions

Currently, there is an increasing need to develop new strategies or combine existing approaches to improve cancer treatment outcomes. This study tested the efficacy of gene electrotransfer of IL-12 in combination with IL-2 in CT26 mouse colon carcinomas. The results confirmed the feasibility, safety, and efficacy of this therapeutic approach also in immunogenic CT26 tumours. Our combined approach resulted in activation of the immune system, which in turn eradicated the tumour and led to immune memory. However, the therapy should be refined to further improve antitumour efficacy.

## Figures and Tables

**Figure 1 ijms-24-12900-f001:**
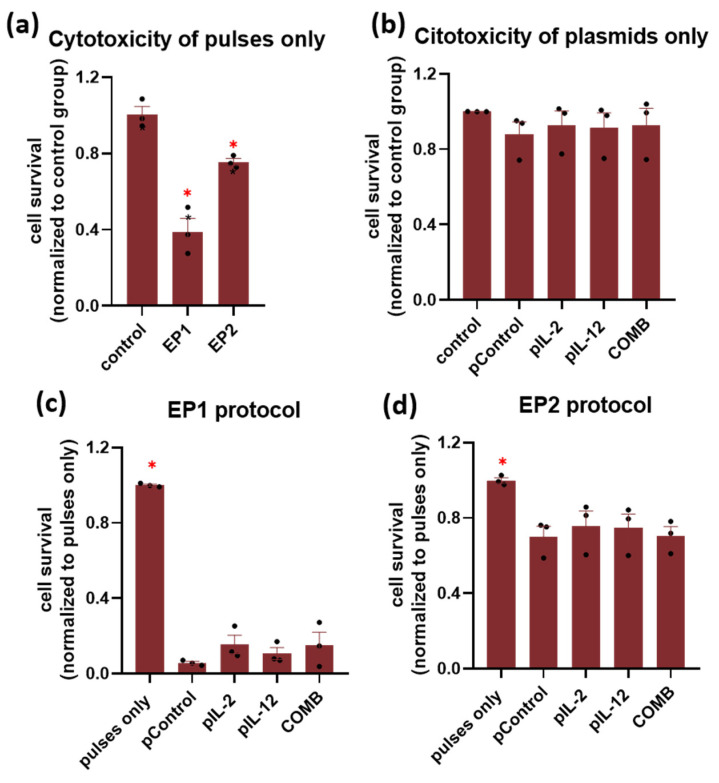
In vitro survival of cells after the treatment with (**a**) two different pulse protocols and (**b**) plasmids alone (pIL-2, pIL-12, and pControl) and (**c**,**d**) after the addition of plasmids coupled with two different pulse protocols in CT26 cells. The results are shown as cell survival normalized to the control group. The experiments were performed in 3 biological replicates, each consisting of 8 technical parallels. Legend: * *p* < 0.05 values were considered statistically significant compared to the control group. The values are expressed as the AM ± SEM.

**Figure 2 ijms-24-12900-f002:**
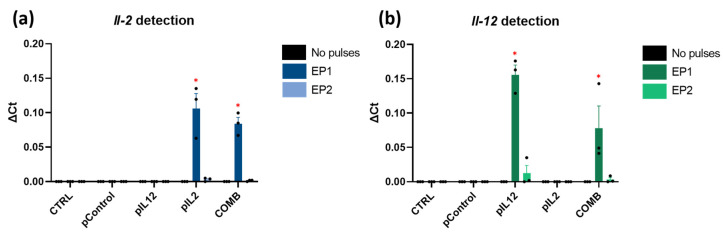
In vitro (**a**) *Il-2* and (**b**) *Il-12* mRNA expression after gene electrotransfer. The data are plotted as ΔCt (the relative expression to the average of the two housekeeping Actb and Gapdh genes). The experiments were performed in 3 biological replicates, each consisting of 2 technical parallels. Legend: * *p* < 0.05 comparing groups treated with EP1 or EP2 pulse protocol. N.D. = not detected. The values are expressed as the AM.

**Figure 3 ijms-24-12900-f003:**
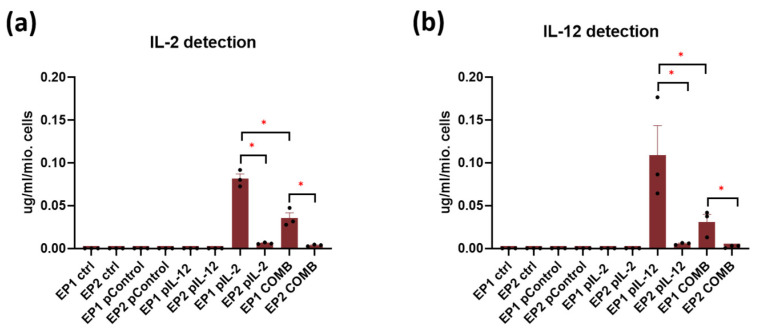
In vitro (**a**) IL-2 and (**b**) IL-12 protein expression after gene electrotransfer. The data are plotted as a concentration of the protein [μg/mL] per million cells. The experiments were performed in 3 biological replicates, each consisting of 4 technical parallels. Legend: * *p* < 0.05 values were considered statistically significant. The values are expressed as the AM ± SEM.

**Figure 4 ijms-24-12900-f004:**
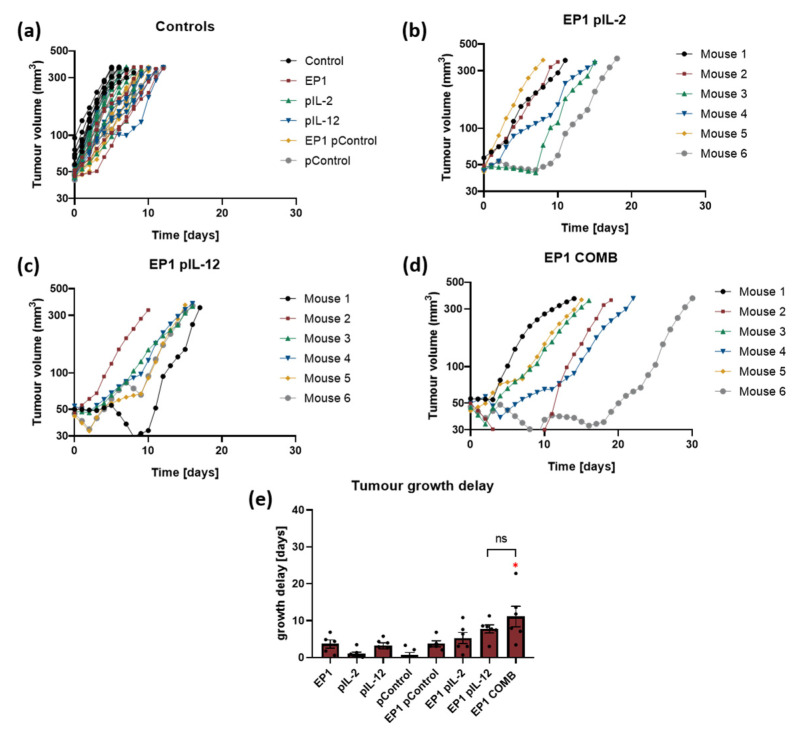
Therapeutic efficacy in murine CT26 colon carcinoma. (**a**) Tumour growth curves of individual tumours in control groups (*n* = 6). (**b**) Tumour growth curves in the EP1 pIL-2 group (*n* = 6), (**c**) EP1 pIL-12 group (*n* = 6), and (**d**) EP1 COMB group (*n* = 6) are shown individually per mouse. (**e**) Tumour growth delay data are normalised to the untreated control group. Legend: * *p* < 0.05 values were considered statistically significant compared to all of the groups except EP1 pIL-12, ns: not statistically significant. The values on graph (**e**) are expressed as the AM ± SEM.

**Figure 5 ijms-24-12900-f005:**
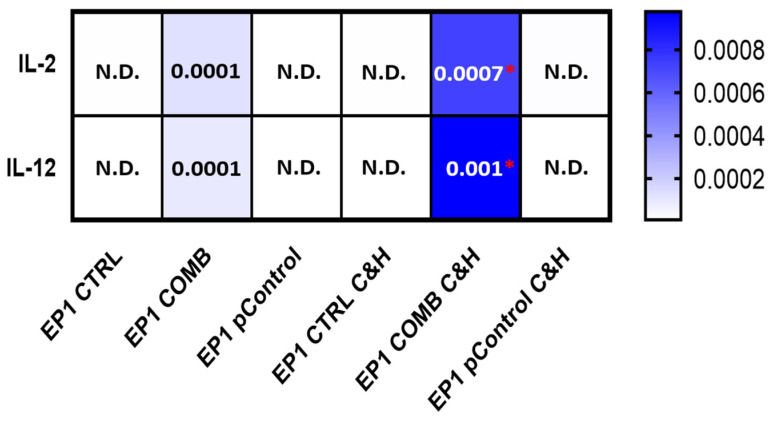
Comparison of *Il-2* and *Il-12* mRNA expression after gene electrotransfer in tumours pre-treated with collagenase and hyaluronidase and without this pre-treatment. The data are plotted as ΔCt (the relative expression to the average of the two housekeeping Actb and Gapdh genes). The experiments were performed in 6 biological replicates, each consisting of 2 technical parallels. Legend: * *p* < 0.05 comparing groups pre-treated with collagenase and hyaluronidase and without the pre-treatment. N.D. = not detected. The values are expressed as the AM.

**Figure 6 ijms-24-12900-f006:**
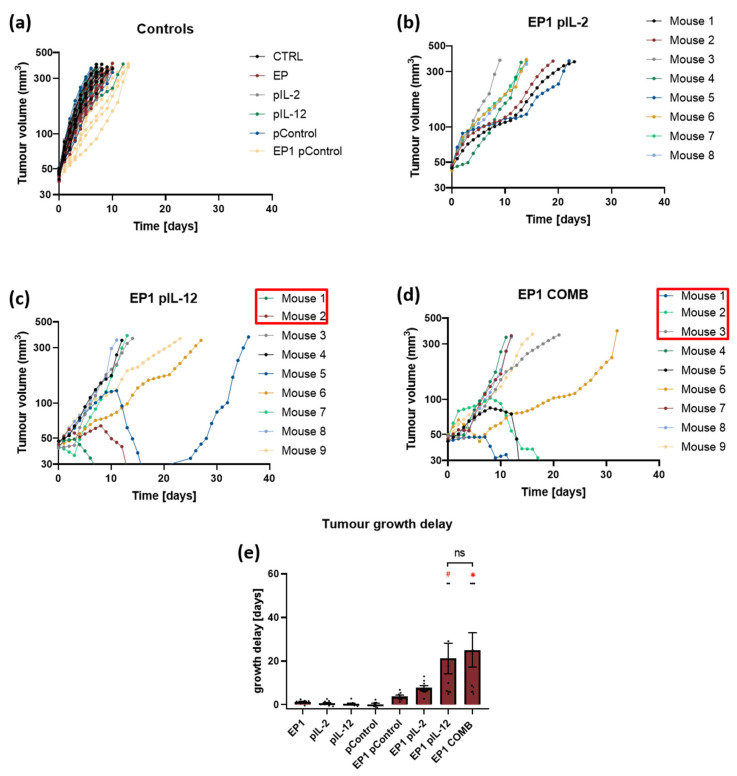
Therapeutic efficacy in murine CT26 colon carcinoma pre-treated with collagenase and hyaluronidase. (**a**) Tumour growth curves of individual tumours in control groups (*n* = 6). Tumour growth curves of (**b**) EP1 pIL-2 (*n* = 8), (**c**) EP1 pIL-12 (*n* = 9), and (**d**) EP1 COMB group (*n* = 9) are shown individually per mouse. (**e**) Tumour growth delay data are normalised to the untreated control group. Mice with complete response (tumour free 100 days after the treatment) are marked with a red square. Legend: * *p* < 0.05 values were considered statistically significant compared to the control group, ns: not statistically significant, # *p* < 0.05 values were considered statistically significant in comparison to electroporated and non-electroporated control groups. The values on graph are expressed as the AM ± SEM.

**Figure 7 ijms-24-12900-f007:**
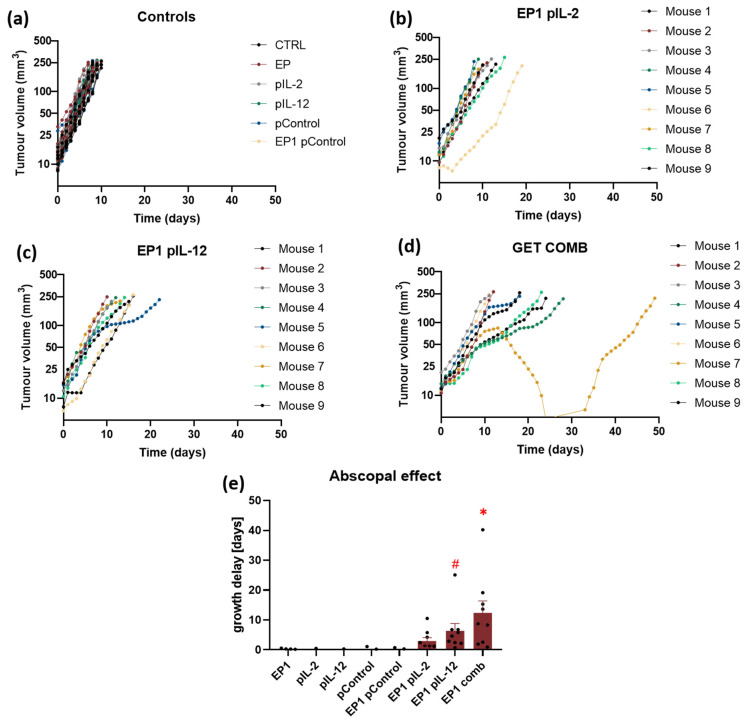
Abscopal effect in murine CT26 colon carcinoma. (**a**) Tumour growth curves of individual tumours in control groups (*n* = 8). Tumour growth curves of (**b**) EP1 pIL-2 group (*n* = 8), (**c**) EP1 pIL-12 group (*n* = 9), and (**d**) EP1 COMB group (*n* = 9) are shown individually per mouse. (**e**) Tumour growth delay data were normalised to the untreated control group. For the secondary tumour, the volume of 200 mm^3^ was set as the human endpoint of the experiment. Legend: * *p* < 0.05 values were considered statistically significant compared to all of the groups. The values are on graph are expressed as the AM ± SEM, # *p* < 0.05 values were considered statistically significant in comparison to electroporated and non-electroporated control groups.

**Figure 8 ijms-24-12900-f008:**
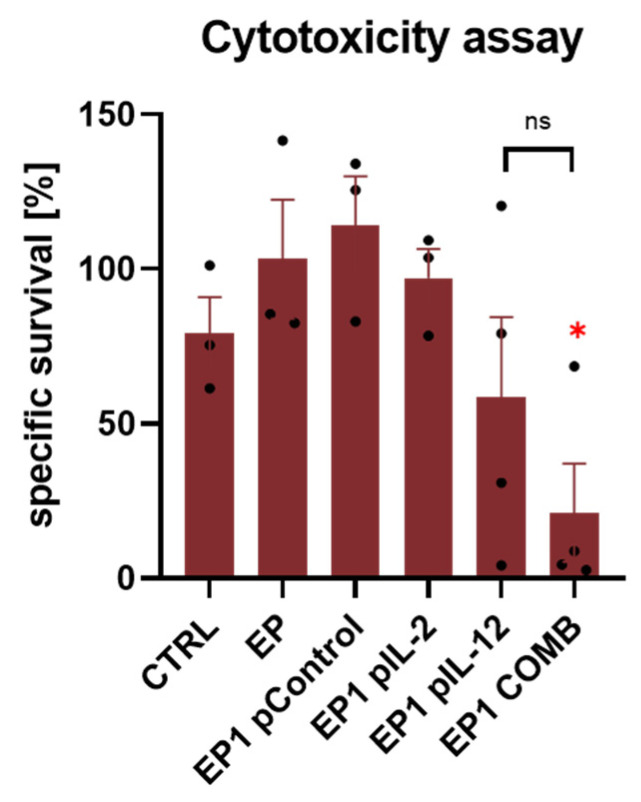
Cytotoxicity assay in murine CT26 colon carcinoma. Specific survival of tumour cells in CTRL (*n* = 3), EP (*n* = 3), EP1 pIL-2 (*n* = 3), EP1 pControl (*n* = 3), EP1 pIL-12 (*n* = 4) and EP1 COMB (*n* = 4) groups. Legend: * *p* < 0.05 values were considered statistically significant, ns: not statistically significant. The values are expressed as the AM ± SEM.

**Figure 9 ijms-24-12900-f009:**
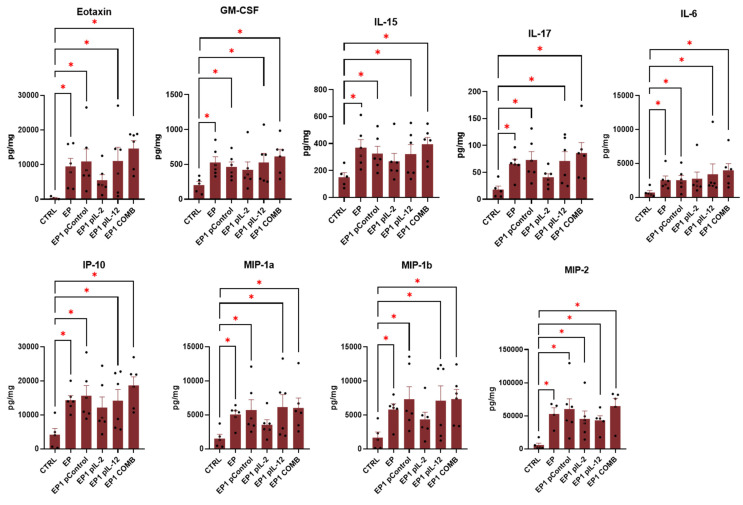
The concentrations of Eotaxin, GM-CSF, IL-15, IL-17, IL-6, IP-10, MIP-1a, MIP-1b, and MIP-2 that were significantly increased in the tumour samples of EP, EP1 pControl, EP1 pIL-2, EP1 pIL-12 and EP1 COMB groups compared to CTRL group (*n* = 6). The concentration of cytokines is presented in pg/mg. Legend: * *p* < 0.05 values were considered statistically significant compared to the marked group. The values are expressed as the AM ± SEM.

**Figure 10 ijms-24-12900-f010:**
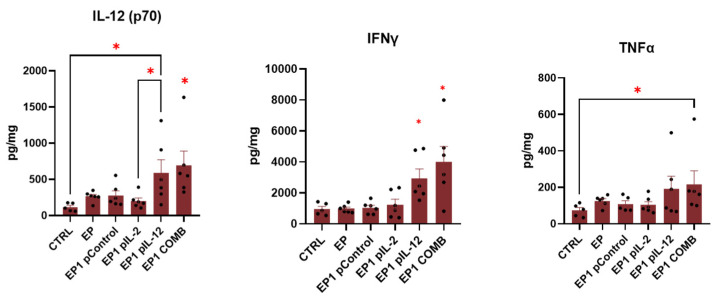
The concentrations of IL-12 (p70), IFN-γ and TNF-α that were significantly increased in the tumour samples of EP1 pIL-12 and EP1 COMB groups in comparison to CTRL group (*n* = 6). The concentration of cytokines is presented in pg/mg. Legend: * *p* < 0.05 values were considered statistically significant compared to the marked or all the other groups. The values are expressed as the AM ± SEM.

**Figure 11 ijms-24-12900-f011:**
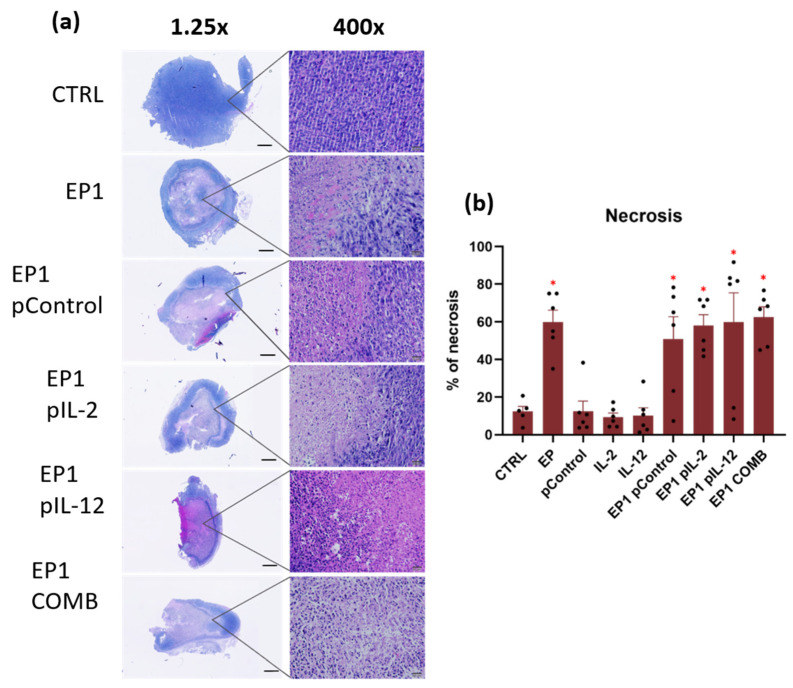
(**a**) Representative images of tumour paraffin sections stained with H&E 3 days after the treatment, 1.25× magnification, scale bar = 1 mm; 400× magnification, scale bar = 20 µm. (**b**) Area of necrosis on tumour sections stained with H&E 3 days after treatment. The values are expressed as the AM ± SEM per entire tumour section; *n* = 6 tumours per group. Legend: * *p* < 0.05 values were considered statistically significant in comparison to the tumour in non-electroporated control group.

**Figure 12 ijms-24-12900-f012:**
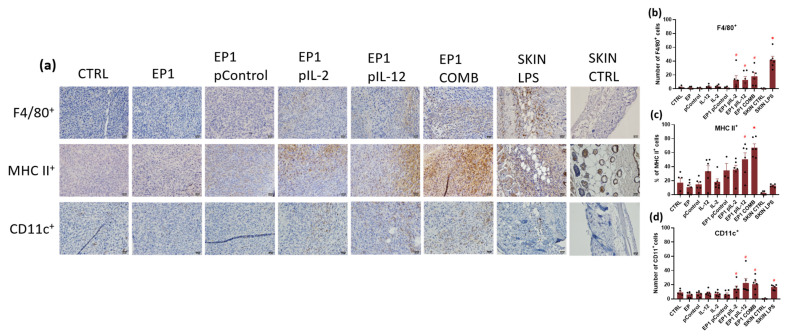
(**a**) Representative images of colon carcinoma paraffin sections stained with anti-F4/80 and anti-MHC II antibody specific for macrophages and M1 macrophages and anti-CD11c antibody specific for dendritic cells that represent the cells of innate immune response. Immunopositive cells are stained brown. The images were taken at 400× magnification, the scale bar corresponds to 20 μm. Graph representing the number of (**b**) F4/80, (**c**) MHC II and (**d**) CD11c positive cells in a field of view. The values are expressed as the AM ± SEM per field of view; *n* = 6 tumours per group, of which at least 4 viable parts of the tumour were analysed. Legend: * *p* < 0.05 values were considered statistically significant in comparison to all of the groups, # *p* < 0.05 values were considered statistically significant in comparison to electroporated and non-electroporated control groups.

**Figure 13 ijms-24-12900-f013:**
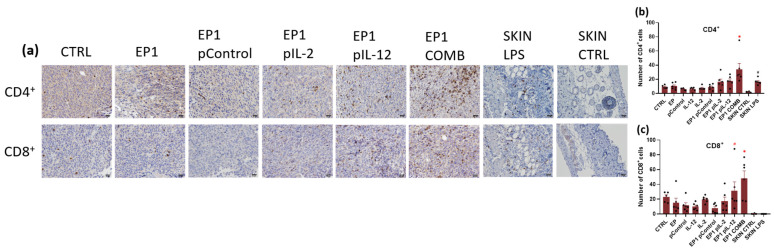
(**a**) Representative images of colon carcinoma paraffin sections stained with anti-CD4 antibody specific for helper T lymphocytes and anti-CD8 antibody specific for cytotoxic T lymphocytes that represent the cells of adaptive immune response. Immunopositive cells are stained brown. The images were taken at 400× magnification, the scale bar corresponds to 20 μm. Graph representing the number of (**b**) CD4 and (**c**) CD8 positive cells in a field of view. The values are expressed as the AM ± SEM per field of view; *n* = 6 tumours per group, of which at least 4 viable parts of the tumour were analysed. Legend: * *p* < 0.05 values were considered statistically significant in comparison to all of the groups, # *p* < 0.05 values were considered statistically significant in comparison to electroporated and non-electroporated control groups.

**Figure 14 ijms-24-12900-f014:**
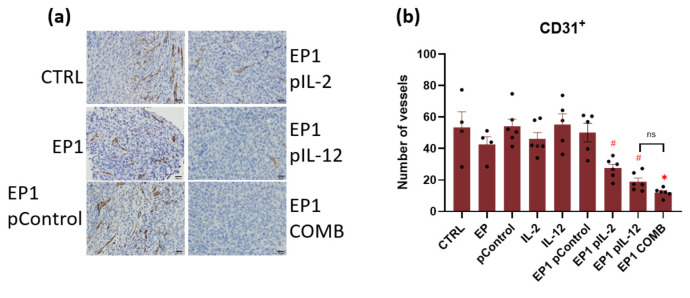
(**a**) Representative images of colon carcinoma paraffin sections stained with anti-CD31 antibody specific for endothelial cells. The positive cells are stained brown. The images were taken at 400× magnification, the scale bar corresponds to 20 μm. Graph representing the average number of vessels in a field of view (**b**). The values are expressed as the AM ± SEM per field of view; *n* = 6 tumours per group, of which at least 4 viable parts of the tumour were analysed. Legend: * *p* < 0.05 values were considered statistically significant in comparison to all of the groups, ns: not statistically significant, # *p* < 0.05 values were considered statistically significant in comparison to electroporated and non-electroporated control groups.

## Data Availability

The data presented in this study are available on request from the corresponding author.

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
