# Peer review of "Gene Immunotherapy of Colon Carcinoma with IL-2 and IL-12 Using Gene Electrotransfer"

_ijms, 2023, doi:10.3390/ijms241612900_

Round 1

Reviewer 1 Report

Article by Komel et al. submitted for review, presents an interesting and current problem of using new methods in oncological therapy. The previous works cited by the authors proove to the authors' extensive experience in this field.

The work is written concisely, logically, despite the extensive experimental apparatus (including cell culture, plasmids, in vivo and in vitro tests), it is easy to read, it does not contain unnecessary repetitions and ambiguities.

With minor comments:

1. In Abstract, it is worth adding the information that the described research concerns an animal model, the experiments were carried out on mice.

2. Line 93 - it is not known what the symbol (?) stands for

3. You need to correct errors like 106 instead of 10^6 (line 116, 149, 150, 174, 220, 238, 258, 259), Il-2 instead of IL-2 (line 158, 160, 873), L instead of mL (line 170 , 172)

4. It is worth to add even a short paragraph at the end of the work on the role of VEGF in various diseases - continuation of line 856; it is possible to quote e.g. doi: 10.5114/ada.2020.95954. "Vascular endothelial growth factor-activated basophils in asthmatics". Progress Dermatol Allergol. 2020 Aug;37(4):584-589.

Congratulations on your work, please include the above comments in the revised version.

Best regards.

 Minor editing of English language required

Reviewer 2 Report

This paper explores the use of gene immunotherapy with IL-2 and IL-12 using gene electrotransfer as a potential treatment for colon carcinoma. The study investigates the effects of immunostimulatory cytokines on immune cells in tumours and presents promising results for future cancer treatments. The data was well organized, and neat presented. Overall, the paper provides valuable insights into the potential of gene immunotherapy for cancer treatment. However, major revision is required to improve the quality of work.

Major revision

1. The author published the "Gene electrotransfer of IL-2 and IL-12 plasmids effectively eradicated murine B16.F10 melanoma" in 2021, and the EP methods in current paper is similar. Since the EP methodology is not novel, the author can consider concise the part and highlight the in vivo data.

2. Global cytotoxicity of IL-2 and IL-12 is the major concern for the immunocytokine treatment, which can be evaluated from mice body weight, ALT/AST measurement and peripheral cytokine profiling.

3. The author needs more efforts to fix the format, in particular the number presenting. For example, "?" at line 93; line 170 and 172 etc. 

The author needs more efforts to fix the format, in particular the number presenting. For example, "?" at line 93; line 170 and 172 etc. 

Round 2

Reviewer 2 Report

Thanks to authors for addressing the comments. Congrats. 

Author Response

Thank you.